# Population-Based Registry Analysis of Antidiabetics Dispensations: Trend Use in Spain between 2015 and 2018 with Reference to Driving

**DOI:** 10.3390/ph13080165

**Published:** 2020-07-25

**Authors:** Eduardo Gutiérrez-Abejón, Paloma Criado-Espegel, Francisco Herrera-Gómez, F. Javier Álvarez

**Affiliations:** 1Pharmacological Big Data Laboratory, Faculty of Medicine, University of Valladolid, 47005 Valladolid, Spain; egutierreza@saludcastillayleon.es (E.G.-A.); alvarez@med.uva.es (F.J.Á.); 2Technical Direction of Pharmaceutical Assistance, Gerencia Regional de Salud de Castilla y León, 47007 Valladolid, Spain; 3Gerencia de Asistencia Sanitaria-Sanidad de Castilla y León, 34001 Palencia, Spain; pcriado@saludcastillayleon.es; 4Nephrology, Hospital Virgen de la Concha—Sanidad de Castilla y León, 49022 Zamora, Spain; 5CEIm, Hospital Clínico Universitario de Valladolid—Sanidad de Castilla y León, 47003 Valladolid, Spain

**Keywords:** antidiabetics, diabetes mellitus, insulins, driving impairing medicines, driving under the influence

## Abstract

Insulins and some oral antidiabetics are considered to be driving-impairing medicines (DIM) and they belong to the Driving under the Influence of Drugs, alcohol, and medicines (DRUID) category I (minor influence on fitness to drive). The trend of antidiabetics use in Castilla y León from 2015 to 2018 is presented through a population-based registry study. Treatment duration with these medicines and the concomitant use of other DIMs were observed. An adjustment method was used with information from the drivers’ license census. For all calculations, age and gender were taken into account. 3.98% of the general population used at least one antidiabetic, as well as 2.92% of drivers. The consumption of antidiabetics in men was higher than in women (4.35% vs. 3.61%, *p* = 0.001), and the use increases with age, especially from 35–39 years to 75–79 years in men and 85–89 years in women. Antidiabetics were consumed chronically, specifically 100% in the case of insulins and 95% in the case of oral antidiabetics. In addition to antidiabetics, 2.5 ± 1.86 DIMs were consumed, mainly anxiolytics (25.53%), opioids (23.03%), other analgesics and antipiretics (19.13%), and antidepressants (17.73%). Collaboration between pharmacists and physicians is a priority to clearly transmitting risks to patients. It is necessary that the health authorities include information on DIMs, such as the DRUID classification, in the prescription and dispensing software.

## 1. Introduction

Diabetes Mellitus (DM) is one of the most prevalent chronic disease worldwide [1], affecting more than 400 million people [2]. In 2019, in Spain, DM affected one in ten adults between 20 and 79 years of age, reaching a prevalence of 10.5%, which is considered to be slightly elevated as compared to the average in European countries (8.9%) [3].

Diabetes can affect fitness to drive [1,4,5,6]. Chronic complications, such as retinopathy or peripheral neuropathy, can impair sensory or motor function [6,7]. On the other hand, acute complications, such as hypoglycemia or hyperglycemia, can affect perception, motor skills, cognition, and judgment, and may cause loss of consciousness during driving [7,8], which may result in road crashes [9,10,11]. Indeed, hypoglycemia is one of the main adverse effects of insulin therapy [10,12,13], as well as of oral antidiabetics, especially sulfonylureas and meglitinides, [12,14]. Notwithstanding, the results from studies on driving impairment in patients with DM may be contradictory, even in the face of the existing evidence on hypoglycemia [15], leading to the current restrictions for these patients to drive, as established in Europe and the USA [4,8,16].

### Study Aim

In Spain, since 2000, DM prevalence has increased, and, thus, the use of insulins (57.5%) and oral antidiabetics (52.5%), which, overall, is 13% higher than in 2000 [17], with a predominance of users of oral antidiabetics [18].

The risks for driving associated with insulin and many oral antidiabetics is well recognized into the Driving Under the Influence of Drugs, alcohol and medicines (DRUID) classification (Appendix A), which group these medicines in the category I (i.e., minor influence on fitness to drive) [19], mainly due to the likelihood of hypoglycemia occurrence during treatment. However, although there are studies on antidiabetic use at national and European level, to our knowledge, no data on the use of these medicines in the driver population are available.

This study presents findings on consumption of insulins and oral antidiabetics in a European population. Data on dispensation at pharmacies of these medicines in the largest region of Spain for the years 2015 to 2018 were assessed. Our analysis also considers the duration of treatment and concomitant use of these medicines with other driving impairing medicines (DIM), and the distribution by age and gender of users. Finally, as previously performed in other studies by our team, an adjustment method has been carried out in order to estimate the real consumption of these medicines in the driver population [20,21,22,23,24].

## 2. Results

From 2015 to 2018, five millions of packages of antidiabetics were dispensed to the population (Appendix A), mostly oral antidiabetics (71.47%) than insulins (28.53%). These findings show that the real proportion of consumers into the general population was 3.98%, 3.13% under an oral antidiabetic, 1.56% under insulin, and 0.71% under both an oral antidiabetic and an insulin. Men were most commonly under treatment with antidiabetics than women (4.35% vs. 3.61%, *X*^2^ = 18831.883, *p* = 0.001; Table 1). As expected, 100% of insulins and more than 95% of oral antidiabetics was of chronic use (Table 1). The consumption of all antidiabetics (Figure 1) and separately insulins and oral antidiabetics increase with age (Figure 2), especially from 35–39 years to 75–79 years in men and to 85–89 years in women.

Sitagliptin in combination with metformin (Anatomical Therapeutic Chemical (ATC) code A10BD07) and vildagliptin in combination with metformin (ATC code A10BD08), were the most consumed oral antidiabetics (more than 335,000 packages per year). Insulin glargine (ATC code A10AE04) and insulin aspart, intermediate or long acting combined with fast acting (ATC code A10AD05) was the most consumed insulins (more than 180,000 packages per year). Detailed oral antidiabetics and insulins dispensation are presented online (Appendix A).

Yearly users of antidiabetics also took 2.50 ± 1.86 DIMs (Table 1), with higher values in women than in men (2.76 vs. 2.19, *t* = −74.63, *p* = 0.001), and represented by anxiolytics (25.53%), opioids (23.03%), other analgesics and antipyretics (19.13%), and antidepressants (17.73%). In addition, chronic use and yearly use of these medicines increased, respectively, by 32.02% (*Z* = 47.25, *p* < 0.0001, 3.31% in 2015 vs. 4.37% in 2018) and 31.55% (*Z* = 36.24, *p* < 0.0001, 3.36% in 2015 vs. 4.42% in 2018; Table 2; Figure 3).

With respect to drivers (Appendix A), 2.92% used at least one antidiabetic, 0.51% taking both an oral antidiabetic and an insulin, being men users four times more frequent than women users (4.22% vs. 0.98%, *X*^2^ = 17,826.133, *p* = 0.001; Table 1) and, as in the general population, in 100% of cases these medicines were used chronically. The consumption increased with age, but, among women drivers, the peak of use of antidiabetics is achieved 20 years earlier than in women into the general population (Figure 1 and Figure 2). Finally, chronic and yearly use of antidiabetics increased by 40% (*Z* = 70.69, *p* < 0.0001, 2.36% in 2015 vs. 3.28% in 2018; Table 2; Figure 3), and the concomitant use of other DIMs was similar to that observed into the general population, with a use of more DIM among women drivers.

## 3. Discussion

According to our results, between 2015 and 2018, almost 4% out of the population and 3% of drivers in Castile and León were under treatment with oral antidiabetics alone or in combination with insulins, and mostly men consumed more antidiabetics than women, both into the general population and among drivers. The chronic use of fixed combinations of oral antidiabetics and insulins were noted, and users took these medicines with more than two driving-impairing medicines, mostly anxiolytics and analgesics, including opioids.

Our findings confirm the increase in the use of antidiabetics claimed by public national data [18], which is consistent with the increase in the global prevalence of DM [3] and with aging of the population, which is particularly relevant in Spain. In the other hand, as compared to other European OECD countries, in Spain the oral antidiabetics and insulin consumption is 14% higher [17]. In this sense, the increase in consumption per age of antidiabetics is also consistent with age distribution of DM, with a lower prevalence in young adults when compared with those more aged [3]. Furthermore, higher use among men is consistent with higher disease prevalence in male sex individuals [3], and higher concomitant use of other driving-impairing medicines is observed among women into both the general population and the driver population (e.g., anxiolytics, opioids, and antidepressants) [21,23,24].

All over the world, the majority of diabetic people have type 2 diabetes (90–95%), which explains the observed greater consumption of oral antidiabetics when compared to insulins [25]. Our results also show that two dipeptidyl peptidase-4 inhibitors (DPP4) considered as second-line treatments were used more frequently, in combination with metformin, probably to induce a greater weight reduction without intense hypoglycemia [26,27]. Moreover, intermediate or long acting insulin preparations were preferred, as expected according to available evidence [28].

Surprisingly, metformin alone that does not include the pictogram “medicines and driving”, which is, not considered a driving-impairing medicine, was the most consumed antidiabetic medicine, with an average of around 715,000 packages per year. According to our findings, metformin was used, on average, by more than 3% of the population each year (45% of the global consumption of oral antidiabetics) [18], which is in line with current recommendations. Nevertheless, hypoglycemia is a patent risk for all antidiabetics. Prudence and common sense is thus recommended to clinicians when prescribing this and other oral antidiabetics that do not include the pictogram “medicines and driving” (Appendix A).

Importantly, the evidence on DM and driving is sometimes conflictive. Koepsell et al. [29] and Cox et al. [30] show an increase in the risk for motor vehicle crashes of around 30% as compared to non-diabetic drivers. However, the Emergency Care Research Institute (ECRI) found an increase in such risk of around 12–19% [31]. It seems that the real risk for car crashes is uncertain or slightly increased: rate ratios in insulin-treated diabetics of 0.54 to 1.8 [25], and standardized incidence ratios in oral antidiabetics-treated and insulin-treated diabetics of 1.2 to 1.4 [5] are found. Indeed, separately, oral antidiabetics and insulins should be associated to a lower risk [10] or no risk [13,32], which suggests that really an adequate monitoring of blood glucose levels of patients, to avoid hypoglycemia is of crucial importance [6]. Therefore, antidiabetics should be prescribed and consumed appropriately, even those not including the “medicines and driving” pictogram. In this sense, susceptible populations, such as chronic kidney disease patients and other individuals affected by chronic and potentially impairing diseases, should be at the focus of personalized strategies to avoid hypoglycemia [33].

In Europe and in the USA, diabetics are restricted for obtaining driver’s licenses [4,8,16,34]. Once more, it is of great importance that healthcare professionals aware of risks for a safe driving in this patient population, in order to provide adequate information to these patients [35], for advising them on risk reduction [30]: an important proportion of healthcare professionals do not know hypoglycemia and its relation for a safe driving [36].

In addition to clinicians, pharmacists take responsibility for educating patients on potential risks for driving through drug counseling [37]. In a study on the implementation of dispensing support tools, it was observed that the inclusion of details on DRUID category of drugs dispensed in the software of daily use had a positive effect in risk communication to patients [38].

Finally, this study has some limitations that should be mentioned. Information on consumption of oral antidiabetics and insulins in hospitals or from the private practice has not been taken into account. Nevertheless, this was not an important impediment, as a high percentage of the population are included in our public health insurance system. Similarly, in the CONCYLIA database there is no information on dispensing “over the counter” medications, but all antidiabetics have a mandatory medical prescription. Lastly, it has been necessary to use an extrapolation method to estimate dispensation of antidiabetics to drivers. For this, weighting was performed to adjust the consumption of antidiabetics among licensed drivers by age and gender, as had made in a previous studies [20,21,22,23,24]. However, the results do not represent an authentical stratification of use, because the CONCYLIA database does not record information on driving.

## 4. Materials and Methods

### 4.1. Real-World Study Details

The results from an epidemiological population-based registry study are presented here in accordance to the REporting of studies Conducted using Observational Routinely-collected Data (RECORD) recommendations, in order to adequately provide real-world evidence into the topic addressed [39]. The years 2015 to 2018 data on dispensations at pharmacies of insulins (ATC subgroups A10AB and A10AC) and many oral antidiabetics (ATC subgroups A10BA, A10BB, A10BD, A10BF, A10BG, A10BH, A10BJ, A10BK, and A10BX), which are considered DIM (Appendix A) were assessed. Since 2011, in our country all DIMs include the “medicines and driving” pictogram on the packaging, with the aim of improving knowledge about the effects of these medicines on driving ability [20], and such pictogram was the way to identify antidiabetics for entering to calculations.

Our pharmaceutical care information system, CONCYLIA (http://www.saludcastillayleon.es/portalmedicamento/es/indicadores-informes/concylia), contains data on the dispensation of all medicines covered by our public health insurance system at pharmacies in Castile and Leon, such as including DIMs. This population registry does not contain information on medicines dispensed at hospitals, into the private practice, and those considered as ‘over the counter’ medications; however, the Spanish national health insurance system cover more than 95% of the population and antidiabetics are only on prescription. There is no recorded information about driving in the CONCYLIA database, so as in other studies carried out by our team [20,21,22,23,24], weighting was performed to obtain the adjusted antidiabetics consumption for licensed drivers according to age and gender while using the Castile and León drivers’ license census data (http://www.dgt.es/es/seguridad-vial/estadisticas-e-indicadores/permisos-conduccion/). Dispensation was considered to be equivalent to consumption, since the adherence rate in Castile and León patients is high. This study was approved by our local ethics committee on 17 March 2016 (reference number PI 16-387).

The following variables were considered: (1) frequency of consumption of antidiabetics, (2) chronic use (≥30 days) and yearly use of antidiabetics, and (3) concomitant use of antidiabetics with other DIMs.

### 4.2. Statistical Analysis

In all analyses, the distribution of the population by age and gender has been taken into account (Appendix A). Frequencies (percentages) with their corresponding 95% confidence interval (95% CI) or as means accompanied by their standard deviations (SD) have been calculated. Differences between continuous variables were calculated while using Student’s *t*-Test (*t*), and those between categorical variables using Pearson’s Chi-squared test (*X*^2^). The Cochran–Armitage trend test was used to evaluate the trends of consumption by years (*Z*). The level of significance was set at *p* ≤ 0.05. All statistical analyzes were performed using the Statistical Package for the Social Sciences (SPSS version 24.0; SPSS Inc, Chicago, IL, USA). Finally, Microsoft Word and Excel (Microsoft Office version 365; Microsoft, Redmon, WA, USA) were used for preparing this manuscript.

## 5. Conclusions

A significant increase in the consumption of oral antidiabetics and insulins has been observed between 2015 and 2018, with a specific pattern of use into the general population and among drivers, and with a concomitant use of other driving-impairing medicines. Despite the fact that the medicines addressed belong to the DRUID category I (minor influence on fitness to drive), the use of antidiabetics not considered as DIM should be taken into account when evaluating risks issues.

The prevention of hypoglycemia is of crucial importance, so that monitoring of patients with DM should be performed [40]. In this sense, healthcare professionals should educate patients with DM on symptoms and the predisposing conditions for this acute complication [14], and promoting adequate control of blood glucose [6]. Evidently, clinicians must particularly be sufficiently prepared and able to decide which people with DM have an unacceptable driving risk and must be excluded from driving [10]. However, in our opinion, health professionals are required to transmit clear and precise information regarding the meaning of the “medicines and driving” pictogram. A Spanish study revealed that only 15.9% of the population knew of the existence of the pictogram and the majority said that they had received no adequate information from the different healthcare professionals concerning the effects of the medicaments on driving abilities [41].

DIMs consumption can be considered a road safety problem, while taking into account that, according to the WHO Global Status Report On Road Safety, traffic collisions cause 1.3 million deaths every year [42], and in Spain at least 34% of the population consumes these medications [20]. Finally, it is necessary to strengthen collaboration between pharmacists and physicians in order to improve the communication of risks to the patient [38], and healthcare authorities should promote the inclusion of additional tools in prescription and dispensing tools.

## Figures and Tables

**Figure 1 pharmaceuticals-13-00165-f001:**
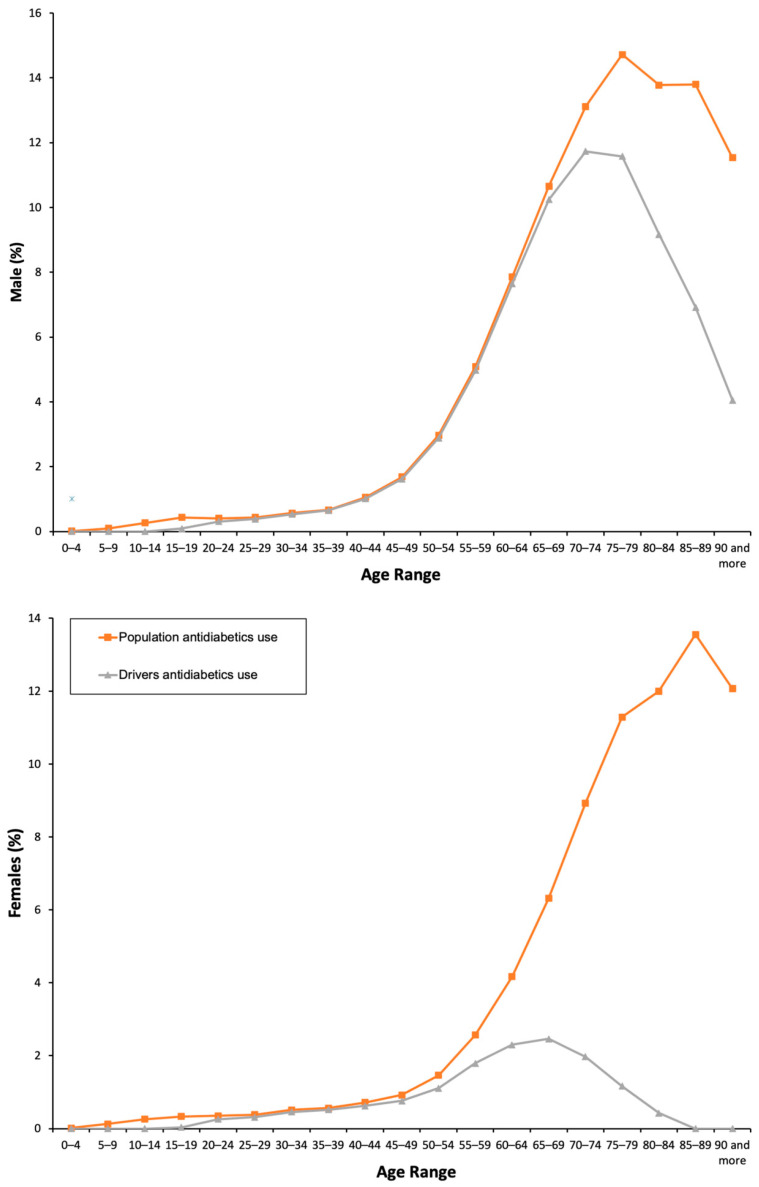
Frequency of antidiabetics use by the general population and the driver population.

**Figure 2 pharmaceuticals-13-00165-f002:**
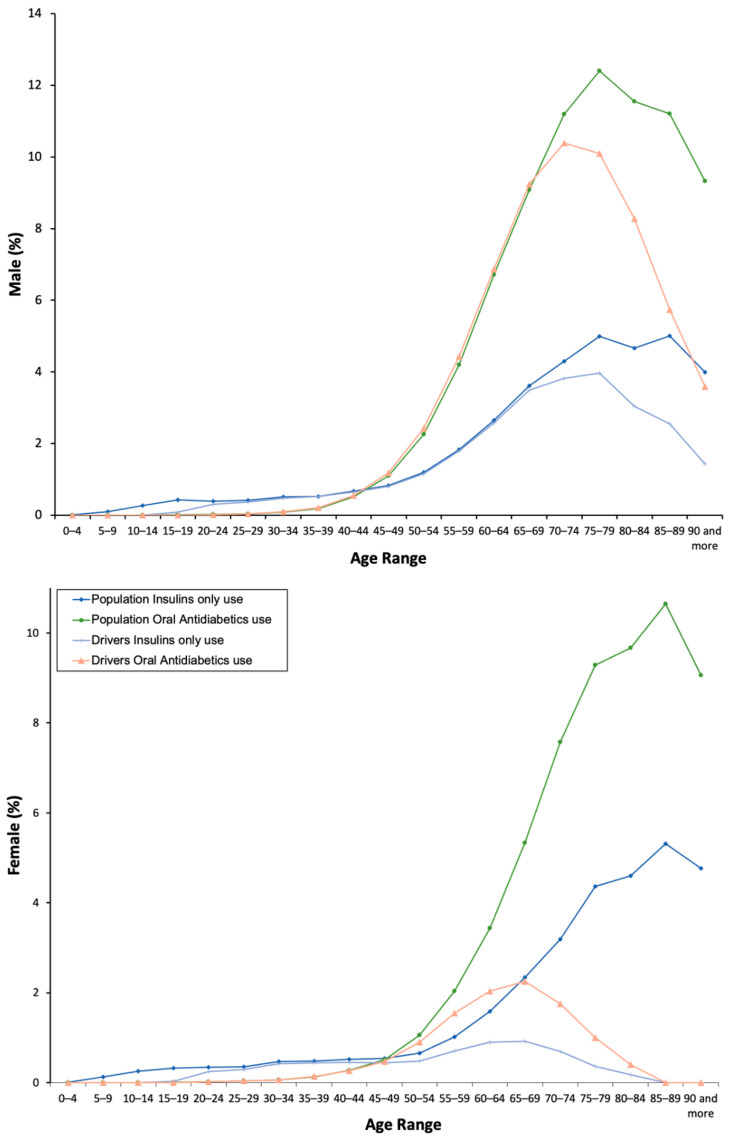
Frequency of use by the type of antidiabetic.

**Figure 3 pharmaceuticals-13-00165-f003:**
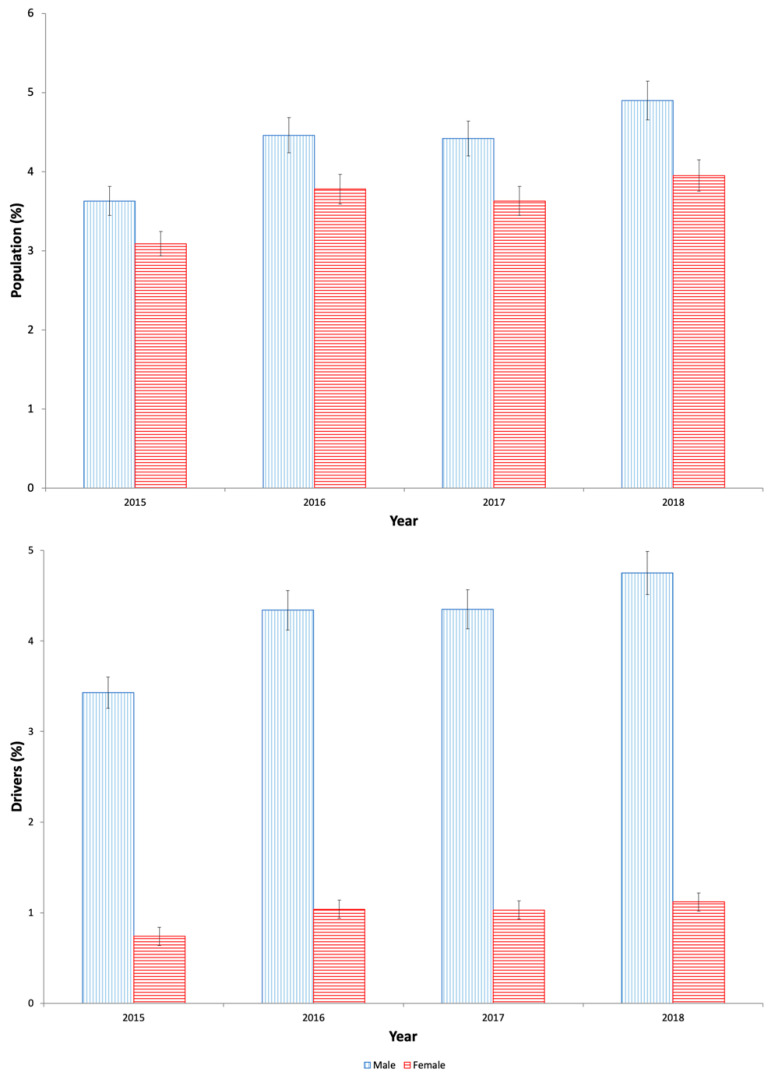
Evolution of antidiabetics use in Castilla y León (2015–2018).

**Table 1 pharmaceuticals-13-00165-t001:** Consumption of antidiabetics according to CONCYLIA database and the Castile and León drivers’ license census data.

Gender	Population Using Antidiabetics % (95CI)			Drivers Using Antidiabetics % (95CI)		
	Total	Insulins	Oral Antidiabetics	Total	Insulins	Oral Antidiabetics
Total	3.98 (3.95–4)	1.56 (1.54–1.58)	3.13 (3.1–3.15)	2.92 (2.89–2.94)	1.15 (1.13–1.16)	2.44 (2.42–2.47)
Male	4.35 (4.32–4.39)	1.64 (1.61–1.66)	3.48 (3.44–3.51)	4.22 (4.17–4.26)	1.58 (1.55–1.61)	3.62 (3.58–3.66)
Female	3.61 (3.58–3.65)	1.49 (1.47–1.51)	2.79 (2.76–2.82)	0.98 (0.96–1.01)	0.5 (0.49–0.52)	0.7 (0.68–0.73)
	*Χ*^2^ = 18,831.883; *p* = 0.001	*Χ*^2^ = 7157.779; *p* = 0.001	*Χ*^2^ = 14,785.013; *p* = 0.001	*Χ*^2^ = 17,826.133; *p* = 0.001	*Χ*^2^ = 8420.777; *p* = 0.001	*Χ*^2^ = 9520.731; *p* = 0.001
Chronic use						
Total	3.93 (3.9–3.95)	1.56 (1.54–1.58)	3.06 (3.04–3.09)	2.88 (2.85–2.91)	1.15 (1.13–1.16)	2.23 (2.21–2.26)
Male	4.3 (4.27–4.34)	1.63 (1.61–1.66)	3.41 (3.38–3.45)	4.17 (4.12–4.21)	1.58 (1.55–1.6)	3.32 (3.28–3.35)
Female	3.57 (3.53–3.6)	1.49 (1.47–1.51)	2.73 (2.7–2.76)	0.97 (0.94–0.99)	0.5 (0.48–0.52)	0.61 (0.59–0.63)
	*Χ*^2^ = 18,734.605; *p* = 0.001	*Χ*^2^ = 7158.431; *p* = 0.001	*Χ*^2^ = 14,639.994; *p* = 0.001	*Χ*^2^ = 17,630.868; *p* = 0.001	*Χ*^2^ = 8420.042; *p* = 0.001	*Χ*^2^ = 9172.203; *p* = 0.001
Average of driving impairing medicines. Population antidiabetics use						
Total	2.5 ± 1.86	2.63 ± 1.95	2.48 ± 1.84	2.25 ± 1.77	2.38 ± 1.88	2.22 ± 1.73
Male	2.19 ± 1.68	2.33 ± 1.79	2.16 ± 1.65	2.17 ± 1.7	2.31 ± 1.81	2.13 ± 1.65
Female	2.76 ± 1.96	2.87 ± 2.04	2.75 ± 1.95	2.68 ± 2.07	2.68 ± 2.11	2.74 ± 2.06
	*t* = −74.63; *p* = 0.001	*t* = −42.45; *p* = 0.001	*t* = −69.96; *p* = 0.001	*t* = −27.53; *p* = 0.001	*t* = −13.22; *p* = 0.001	*t* = −27.94; *p* = 0.001

Abbreviations: 95CI, confidence interval. *Χ*^2^, *t*: Chi squared and *T*-Student test for comparison between men and women.

**Table 2 pharmaceuticals-13-00165-t002:** Evolution of antidiabetics use in Castile and León (2015–2018).

Gender	Population Using Antidiabetics % (95CI)				Drivers Using Antidiabetics % (95CI)			
	2015	2016	2017	2018	2015	2016	2017	2018
Total	3.36 (3.33–3.38)	4.11 (4.09–4.14)	4.02 (3.99–4.04)	4.42 (4.39–4.45)	2.36 (2.34–2.39)	3.02 (2.99–3.05)	3.01 (2.98–3.04)	3.28 (3.25–3.31)
Male	3.63 (3.6–3.67)	4.46 (4.42–4.5)	4.42 (4.39–4.46)	4.9 (4.86–4.94)	3.43 (3.39–3.47)	4.34 (4.3–4.38)	4.35 (4.3–4.39)	4.75 (4.7–4.79)
Female	3.09 (3.06–3.12)	3.78 (3.74–3.81)	3.63 (3.6–3.66)	3.95 (3.92–3.99)	0.74 (0.72–0.76)	1.04 (1.01–1.06)	1.03 (1–1.05)	1.12 (1.1–1.15)
	*Χ*^2^ = 4070.524; *p* = 0.001	*Χ*^2^ = 4863.585; *p* = 0.001	*Χ*^2^ = 4741.491; *p* = 0.001	*Χ*^2^ = 5184.767; *p* = 0.001	*Χ*^2^ = 4160.909; *p* = 0.001	*Χ*^2^ = 4389.486; *p* = 0.001	*Χ*^2^ = 4570.11; *p* = 0.001	*Χ*^2^ = 4805.053; *p* = 0.001
Type of antidiabetic								
Insulins								
Total	1.31 (1.29–1.32)	1.65 (1.64–1.67)	1.57 (1.55–1.58)	1.72 (1.7–1.74)	0.95 (0.94–0.97)	1.21 (1.19–1.23)	1.17 (1.15–1.18)	1.26 (1.24–1.28)
Male	1.36 (1.34–1.38)	1.72 (1.69–1.74)	1.65 (1.62–1.67)	1.82 (1.79–1.84)	1.3 (1.28–1.33)	1.66 (1.64–1.69)	1.61 (1.58–1.63)	1.74 (1.71–1.77)
Female	1.26 (1.24–1.28)	1.59 (1.57–1.61)	1.49 (1.46–1.51)	1.62 (1.6–1.64)	0.42 (0.41–0.44)	0.52 (0.51–0.54)	0.51 (0.49–0.53)	0.56 (0.54–0.58)
	*Χ*^2^ = 1518.236; *p* = 0.001	*Χ*^2^ = 1988.255; *p* = 0.001	*Χ*^2^ = 1785.798; *p* = 0.001	*Χ*^2^ = 1898.464; *p* = 0.001	*Χ*^2^ = 1911.783; *p* = 0.001	*Χ*^2^ = 2146.022; *p* = 0.001	*Χ*^2^ = 2202.878; *p* = 0.001	*Χ*^2^ = 2200.603; *p* = 0.001
Oral antidiabetic								
Total	2.59 (2.57–2.61)	3.19 (3.16–3.21)	3.19 (3.16–3.21)	3.55 (3.52–3.57)	1.78 (1.76–1.8)	2.33 (2.31–2.36)	2.38 (2.35–2.4)	2.62 (2.59–2.65)
Male	2.84 (2.81–2.87)	3.51 (3.48–3.54)	3.56 (3.53–3.6)	4 (3.97–4.04)	2.67 (2.63–2.7)	3.43 (3.4–3.47)	3.52 (3.48–3.56)	3.9 (3.85–3.94)
Female	2.34 (2.32–2.37)	2.87 (2.84–2.9)	2.82 (2.8–2.85)	3.11 (3.08–3.14)	0.42 (0.41–0.44)	0.68 (0.66–0.7)	0.68 (0.66–0.7)	0.75 (0.73–0.77)
	*Χ*^2^ = 3198.26; *p* = 0.001	*Χ*^2^ = 3687.492; *p* = 0.001	*Χ*^2^ = 3749.531; *p* = 0.001	*Χ*^2^ = 4167.533; *p* = 0.001	*Χ*^2^ = 1910.429; *p* = 0.001	*Χ*^2^ = 2355.727; *p* = 0.001	*Χ*^2^ = 2541.26; *p* = 0.001	*Χ*^2^ = 2723.007; *p* = 0.001
Average of driving impairing medicines; Population antidiabetics use								
Total	2.5 ± 1.88	2.53 ± 1.89	2.47 ± 1.84	2.51 ± 1.83	2.22 ± 1.75	2.28 ± 1.81	2.23 ± 1.75	2.28 ± 1.76
Male	2.16 ± 1.66	2.23 ± 1.73	2.16 ± 1.66	2.22 ± 1.68	2.14 ± 1.68	2.2 ± 1.74	2.14 ± 1.67	2.2 ± 1.69
Female	2.77 ± 1.99	2.78 ± 1.98	2.73 ± 1.94	2.75 ± 1.92	2.69 ± 2.09	2.68 ± 2.09	2.67 ± 2.08	2.69 ± 2.02
	*t* = −37.42; *p* = 0.001	*t* = −36.63; *p* = 0.001	*t* = −37.99; *p* = 0.001	*t* = −37.53; *p* = 0.001	*t* = −12.4; *p* = 0.001	*t* = −13.15; *p* = 0.001	*t* = −14.61; *p* = 0.001	*t* = −14.77; *p* = 0.001

Abbreviations: 95CI; confidence interval. *Χ*^2^, *t*: Chi squared and *T*-Student test for comparison between men and women.

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
