# Peer review of "Population-Based Registry Analysis of Antidiabetics Dispensations: Trend Use in Spain between 2015 and 2018 with Reference to Driving"

_pharmaceuticals, 2020, doi:10.3390/ph13080165_

Round 1

Reviewer 1 Report

This paper on trend analysis of antidiabetic medication provides some insight into their use in Spain and changes over time. 

It is not clear how the authors determined the number of drivers actually using these medications – more detail is needed in Methods since most readers are unlikely to read Spanish.  Given possible assumptions and extrapolations into driver use some more detail in the limitations section is required. 

The analysis shows that if any less drivers use these drugs which is likely given that some may not drive because of their diabetes; however to properly assess the effect of these drugs on driving ability linkage to a crash database would at least be needed.

How does this Spanish data compare with other countries?  This would be useful to discuss.

The CONCYLIA link did not work!

The following lines have poor sentence structure requiring revision of these passages, lines 147, 181, 202 and 208 at least.

Author Response

Reviewer 1

This paper on trend analysis of antidiabetic medication provides some insight into their use in Spain and changes over time. 

Point 1. It is not clear how the authors determined the number of drivers actually using these medications – more detail is needed in Methods since most readers are unlikely to read Spanish – Given possible assumptions and extrapolations into driver use some more detail in the limitations section is required.

Response 1: We thank the reviewer for these comments. In this regard, we agree to make the appropriate clarifications. Therefore, the following changes are made in the “Material and Methods” section:

Lines 215-222: The following text has been removed: “As performed previously, consumption into the driver population has been estimated for age and gender by accessing to the information contained in the Castile and Leon drivers' license census (http://www.dgt.es/es/seguridad-vial/estadisticas-e-indicadores/permisos-conduccion/) [20–24], and replaced by: “There is no recorded information about driving in the CONCYLIA database, so as in other studies carried out by our team [20-24], weighting was performed to obtain the adjusted antidiabetics consumption for licensed drivers according to age and gender using the Castile and León drivers’ license census data (http://www.dgt.es/es/seguridad-vial/estadisticas-e-indicadores/permisos-conduccion/).

On the other hand, the following changes are made in the “Discussion” section:

Lines 169-175: The following text has been removed: “Lastly, to estimate dispensation of antidiabetics to drivers, it has been necessary to use an extrapolation method, as information on licensed drivers in Castile and León are not included in pharmaceutical care software [20–24]”, and replaced by: “Lastly, to estimate dispensation of antidiabetics to drivers, it has been necessary to use an extrapolation method. For this, weighting was performed to adjust the consumption of antidiabetics among licensed drivers by age and gender, as had made in a previous studies [20-24]. However, the results don´t represent an authentical stratification of use because the CONCYLIA database doesn´t record information on driving”.

Point 2. The analysis shows that if any less drivers use these drugs which is likely given that some may not drive because of their diabetes; however to properly assess the effect of these drugs on driving ability linkage to a crash database would at least be needed.

Response 2: We agree with the reviewer about the importance of having a traffic crash database that records the accident victim's illness. Unfortunately, we do not have this information in our country. However, you can consult the traffic accidents that occurred in a specific year at this link: http://www.dgt.es/es/seguridad-vial/estadisticas-e-indicadores/ficheros-microdatos-accidentalidad/

The risk of accidents in patients consuming oral antidiabetics and insulins has been extracted from the studies that are named throughout the article, especially in the “Discussion” section.

Point 3. How does this Spanish data compare with other countries?  This would be useful to discuss.

Response 3: We agree with the reviewer's appreciation regarding the usefulness of comparing our results with obtained in other countries. Therefore, the following text is included in the “Discussion” section:

Lines 119-120 : “In the other hand, compared to other European OECD countries, in Spain the oral antidiabetics and insulin consumption is 14% higher [17]”.

Point 4. The CONCYLIA link did not work!

Response 4. Thanks to the reviewer for this note. The link has been modified, and now it works correctly

Line 207: http://www.saludcastillayleon.es/portalmedicamento/es/indicadores-informes/concylia

Point 5. The following lines have poor sentence structure requiring revision of these passages, lines 147, 181, 202 and 208 at least.

Response 5. Thanks to the reviewer for this note. Reviewing the document, the following changes have been made:

Lines 149-152: The following text has been removed: “Once more, our results invite to be cautious as all antidiabetics (even those not including the pictogram "medicines and driving") should be prescribed and consumed appropriately”, and replaced by: “Therefore, antidiabetics should be prescribed and consumed appropriately, even those not including the "medicines and driving" pictogram”.

Lines 209-212: The following text has been removed: “However, this population registry does not contain information on medicines dispensed at hospitals, into the private practice, and those considered as 'over the counter' medications, accounting for less than 5% of the total medicines used by the population”, and replaced by: “This population registry does not contain information on medicines dispensed at hospitals, into the private practice, and those considered as 'over the counter' medications, however the Spanish national health insurance system cover more than 95% of the population and antidiabetics are on prescription only.”

Lines 222-223: The following text has been removed: “Dispensation were took equivalent to consumption by considering higher rates of adherence to pharmacological treatments of the patients in Castile and Leon”, and replaced by: “Dispensation was considered equivalent to consumption since the adherence rate in Castile and León patients is high”

Lines 251-252: The following text has been removed: “However, in our opinion, transmission of clear and precise information about the meaning of the pictogram “medicines and driving” on antidiabetics is also to consider”, and replaced by: “However, in our opinion, health professionals are required to transmit clear and precise information about the meaning of the "medicines and driving" pictogram.”

Reviewer 2 Report

The authors describe an assessment of the use of antidiabetic medication in the principle region of Spain (Castile and Leon) for the years 2015-2018 based on the general dispensing records from pharmacies (public health insurance system), as well as driving license data from the license census. The data specifically evaluates the age of patients consuming the medicines, their gender, the differences between genders, and the use of other ‘driving impairing medicines’ (DIMS) using the DRiving Under the Influence of Drugs, alcohol and medicines (DRUID) classification.

This is an important paper as it draws attention to the increasing use of antidiabetics and their DIMS potential, and the fact that these may be taken concomitantly with other DIMS related medication (without including any social or illicit drug use). The potential use of warning pictograms is also discussed, and how this may benefit patient awareness as well as how pharmacists or other health professionals might use this system to help educate the wider patient public as the potential dangers of not only their condition, but the consequences of taking medications to alleviate or control their chronic conditions.  This is particularly pertinent for diabetes where careful monitoring of blood glucose levels is required to aid appropriate medication use.  Medication h=can result in hypoglycaemia, which can then be classified as ‘DIMS’. The authors have published other related papers and demonstrated a useful methodology for analysing and presenting relevant data, and are to be encouraged in their endeavours to continue this work and draw to the attention of both health professionals and the wider public the significance of driving impaired. Given that road traffic accidents are a leading cause of mortality with 1.35 million deaths recorded in 2016, and ranked amongst the top 10 causes of mortality worldwide (World Health Organization. Global Status Report on Road Safety; World Health Organization: Geneva, Switzerland, 2018).

This paper is of relevance to readers of the journal, and the general findings are of importance to other health professionals the wider public as well.

Some suggested changes that may improve the paper are included in the comments given below.

Page 1

Abstract

Page 1. L8: Is it the usual journal format to include the chi squared value, or other -statistical reference value in the abstract – would a P value alone be sufficient?

 ‘The consumption of antidiabetics in men was higher than in women (4.35% vs. 3.61%, χ² = 27 18831,883, p = 0.001),’

Page 2. L48-51: ‘Notwithstanding, results from studies on driving impairment in patients with DM may be contradictory, even face of the existing evidence on hypoglycemia [15], leading to the current restrictions for these patients to drive established in Europe and the USA [4,8,16].’

Can this be better written as: ‘….even in the face of the existing evidence to drive as established in Europe and the USA [4,8,16].…’

L56-67: ‘The risks for driving associated to with insulin and many oral antidiabetics is well recognized into 56 the DRiving Under the Influence of Drugs, alcohol and medicines (DRUID) classification’

L59-61: ‘mainly due to the likelihood of hypoglycemia occurrence during treatment with. However, although there are studies on antidiabetics use at national and European level, to our 60 knowledge no data on the use of these medicines in the driver population are available.

L62-64: This study presents findings on consumption of insulins and oral antidiabetics in a European population. Data on dispensation at pharmacies of these medicines in the largest region of Spain for the years 2015 to 2018 were assessed. Our analysis considers also the duration of treatment and the concomitant 64 use of these medicines with other driving impairing medicines (DIM),’

Please note: This reviewer (unfortunately does not have the time, noting 7 suggested edits in less than 20 lines, and) will not provide suggestions for further minor grammatical edits that will improve the reading and understanding of the manuscript – but would suggest that the authors/others find a fluent English speaker to read through and make appropriate suggestions (and perhaps may then be acknowledged in the acknowledgment section).

L85: ‘Yearly users of antidiabetics also took 2.50 ± 1.86 DIMs’ – consider inserting ‘also’, to emphasise the point that you have made earlier  - not only are many antidiabetics (Supplementary Table 1) and the sequelae of their consumption associated with impaired driving (e.g. hypoglycaemia), but then diabetic drivers may in addition be consuming other DIMS related medication.

Pages 2 and 3. L 94-96: ‘The consumption increased with age, but among women drivers the peak of use of antidiabetics is achieved 20 years earlier than in women into the (Figures 1 and 2).’ Should this read ‘than in men’ (Figures 1 and 2).’?  From looking at Figures 1. And 2. The peak for Men (male drivers) is around 75-79 years, whilst for women (female drivers) it is around 65-69 years. Assuming this reviewer is interpreting the graphs and their scales correctly – then this would be around 10 years difference, not 20 years?

Page 4 and 5.. Tables 1. And 2. Please add that the statistical comparisons (chi squared and t tests) relate to comparting/contrasting males with females – either in the titles or in the legends. This will help the reader understand what the reported statistics relate to.

Discussion (Please note the ‘official’ page numbers given on the PDF of the manuscript do not align with the actual sequential page numbers. Consequently, manuscript line numbers provide an accurate reference for the reviewers comments).

L121-124: Furthermore, higher use among men is consistent with higher disease prevalence in male sex individuals [3], and higher concomitant use of other driving-impairing medicines is observed among women into both the general population and the driver population (e.g., anxiolytics, opioids, antidepressants) [21,23,24]. Consider adding ‘higher’?

L149-151: ‘other individuals affected by chronic and invalidating diseases should be at the focus of personalized strategies to avoid hypoglycemia [33].’ What is the meaning of invalidating? Should this be ‘chronic and potentially impairing diseases’? Are you referring to ‘invlidating’ their fitness to drive? It is not clear what the meaning is.

Discussion or Conclusion – general point.  Would it be useful to provide context for this important data and interpretation? You might include information on the general level of road traffic accidents worldwide (those killed and seriously injured), as well as known figures for those taking medications/DIMS related medications.  This will help flag up the importance of alerting both help professionals and the general public/patients to the presence and consequences of DIMS driving – and that may be linked to the direct symptoms of their condition or the indirect consequences of medication use (e.g. for antidiabetics - hypoglycemia?

Materials and Methods

L188-228. Would this section be better placed after the introduction and before the results? Thi is the conventional format for reporting studies in journals.

Author Response

Reviewer 2

The authors describe an assessment of the use of antidiabetic medication in the principle region of Spain (Castile and Leon) for the years 2015-2018 based on the general dispensing records from pharmacies (public health insurance system), as well as driving license data from the license census. The data specifically evaluates the age of patients consuming the medicines, their gender, the differences between genders, and the use of other ‘driving impairing medicines’ (DIMS) using the DRiving Under the Influence of Drugs, alcohol and medicines (DRUID) classification.

This is an important paper as it draws attention to the increasing use of antidiabetics and their DIMS potential, and the fact that these may be taken concomitantly with other DIMS related medication (without including any social or illicit drug use). The potential use of warning pictograms is also discussed, and how this may benefit patient awareness as well as how pharmacists or other health professionals might use this system to help educate the wider patient public as the potential dangers of not only their condition, but the consequences of taking medications to alleviate or control their chronic conditions.  This is particularly pertinent for diabetes where careful monitoring of blood glucose levels is required to aid appropriate medication use.  Medication h=can result in hypoglycaemia, which can then be classified as ‘DIMS’. The authors have published other related papers and demonstrated a useful methodology for analysing and presenting relevant data, and are to be encouraged in their endeavours to continue this work and draw to the attention of both health professionals and the wider public the significance of driving impaired. Given that road traffic accidents are a leading cause of mortality with 1.35 million deaths recorded in 2016, and ranked amongst the top 10 causes of mortality worldwide (World Health Organization. Global Status Report on Road Safety; World Health Organization: Geneva, Switzerland, 2018).

This paper is of relevance to readers of the journal, and the general findings are of importance to other health professionals the wider public as well.

Some suggested changes that may improve the paper are included in the comments given below.

Point 1. Abstract. Page 1. L8: Is it the usual journal format to include the chi squared value, or other -statistical reference value in the abstract – would a P value alone be sufficient? ‘The consumption of antidiabetics in men was higher than in women (4.35% vs. 3.61%, χ² = 27 18831,883, p = 0.001),’

Response 1: Thanks for the appreciation. There is no note about it in the “Instructions for authors”. However, the reviewer is correct and it is not necessary to include the Chi squared value in the abstract.

Lines 27-28: “The consumption of antidiabetics in men was higher than in women (4.35% vs. 3.61%, p = 0.001),”

Point 2. L48-51: ‘Notwithstanding, results from studies on driving impairment in patients with DM may be contradictory, even face of the existing evidence on hypoglycemia [15], leading to the current restrictions for these patients to drive established in Europe and the USA [4,8,16].’

Can this be better written as: ‘….even in the face of the existing evidence to drive as established in Europe and the USA [4,8,16].…’

Response 2: changes suggested by the reviewer are made.

Lines 48-51: “Notwithstanding, results from studies on driving impairment in patients with DM may be contradictory, even in the face of the existing evidence on hypoglycemia [15], leading to the current restrictions for these patients to drive as established in Europe and the USA [4,8,16].”

Point 3. L56-57: ‘The risks for driving associated to with insulin and many oral antidiabetics is well recognized into 56 the DRiving Under the Influence of Drugs, alcohol and medicines (DRUID) classification’

Response 3: changes suggested by the reviewer are made.

Lines 56-57: “The risks for driving associated with insulin and many oral antidiabetics is well recognized into the DRiving Under the Influence of Drugs, alcohol and medicines (DRUID) classification…”

Point 4. L59-61: ‘mainly due to the likelihood of hypoglycemia occurrence during treatment with. However, although there are studies on antidiabetics use at national and European level, to our 60 knowledge no data on the use of these medicines in the driver population are available.

Response 4: changes suggested by the reviewer are made.

Lines 59-61: “…mainly due to the likelihood of hypoglycemia occurrence during treatment. However, although there are studies on antidiabetic use at national and European level, to our knowledge no data on the use of these medicines in the driver population are available”

Point 5. L62-64: This study presents findings on consumption of insulins and oral antidiabetics in a European population. Data on dispensation at pharmacies of these medicines in the largest region of Spain for the years 2015 to 2018 were assessed. Our analysis considers also the duration of treatment and the concomitant 64 use of these medicines with other driving impairing medicines (DIM),’

Response 5: changes suggested by the reviewer are made.

Lines 62-64: “This study presents findings on consumption of insulins and oral antidiabetics in a European population. Data on dispensation at pharmacies of these medicines in the largest region of Spain for the years 2015 to 2018 were assessed. Our analysis considers also the duration of treatment and the concomitant use of these medicines with other driving impairing medicines (DIM),”

Please note: This reviewer (unfortunately does not have the time, noting 7 suggested edits in less than 20 lines, and) will not provide suggestions for further minor grammatical edits that will improve the reading and understanding of the manuscript – but would suggest that the authors/others find a fluent English speaker to read through and make appropriate suggestions (and perhaps may then be acknowledged in the acknowledgment section).

We agree with the reviewer's suggestion, and the text has been revised.

Point 6. L85: ‘Yearly users of antidiabetics also took 2.50 ± 1.86 DIMs’ – consider inserting ‘also’, to emphasise the point that you have made earlier  - not only are many antidiabetics (Supplementary Table 1) and the sequelae of their consumption associated with impaired driving (e.g. hypoglycaemia), but then diabetic drivers may in addition be consuming other DIMS related medication.

Response 6: Thanks to the reviewer for this note. Suggested modification is made.

Line 85: “Yearly users of antidiabetics also took 2.50 ± 1.86 DIMs

Point 7. L 94-96: ‘The consumption increased with age, but among women drivers the peak of use of antidiabetics is achieved 20 years earlier than in women into the (Figures 1 and 2).’ Should this read ‘than in men’ (Figures 1 and 2).’?  From looking at Figures 1. And 2. The peak for Men (male drivers) is around 75-79 years, whilst for women (female drivers) it is around 65-69 years. Assuming this reviewer is interpreting the graphs and their scales correctly – then this would be around 10 years difference, not 20 years?

Response 7: In this case, reference is made to the peak consumption of women drivers (65-69) with respect women in the general population (85-89). With this comparison, it is clear that the number of female drivers is lower after a certain age group.

Point 8. Page 4 and 5.. Tables 1. And 2. Please add that the statistical comparisons (chi squared and t tests) relate to comparting/contrasting males with females – either in the titles or in the legends. This will help the reader understand what the reported statistics relate to.

Response 8: We agree with the reviewer's suggestion. as a consequence, the following text has been added in the legends of both tables: “Χ², t: Chi squared and T-Student test for comparison between men and women.”

Point 9. L121-124: Furthermore, higher use among men is consistent with higher disease prevalence in male sex individuals [3], and higher concomitant use of other driving-impairing medicines is observed among women into both the general population and the driver population (e.g., anxiolytics, opioids, antidepressants) [21,23,24]. Consider adding ‘higher’?

Response 9: Thanks to the reviewer for this note. Suggested modification is made.

Lines 122-126: “Furthermore, higher use among men is consistent with higher disease prevalence in male sex individuals [3], and higher concomitant use of other driving-impairing medicines is observed among women into both the general population and the driver population (e.g.,anxiolytics, opioids, antidepressants) [21,23,24]”

Point 10. L149-151: ‘other individuals affected by chronic and invalidating diseases should be at the focus of personalized strategies to avoid hypoglycemia [33].’ What is the meaning of invalidating? Should this be ‘chronic and potentially impairing diseases’? Are you referring to ‘invlidating’ their fitness to drive? It is not clear what the meaning is.

Response 10: Again, thanks to the reviewer for this note. We agree that invalidating is not entirely correct in this context, so the text is modified:

Lines 153-154: “other individuals affected by chronic and potentially impairing diseases should beat the focus of personalized strategies to avoid hypoglycemia [33].”

Point 11. Discussion or Conclusion – general point.  Would it be useful to provide context for this important data and interpretation? You might include information on the general level of road traffic accidents worldwide (those killed and seriously injured), as well as known figures for those taking medications/DIMS related medications.  This will help flag up the importance of alerting both help professionals and the general public/patients to the presence and consequences of DIMS driving – and that may be linked to the direct symptoms of their condition or the indirect consequences of medication use (e.g. for antidiabetics - hypoglycemia?

Response 11. Thanks to the reviewer for this suggestion. A series of global data regarding the consumption of DIMs and deaths from traffic collisions has been included in the “Conclusions” section. In addition, reference 42 has been added:

Lines 259-261: “DIMs consumption can be considered a road safety problem, taking into account that, according to the WHO Global Status Report On Road Safety, traffic collisions cause 1.3 million deaths every year [42], and in Spain at least 34% of the population consumes these medications [20].”

Point 12. L188-228. Would this section be better placed after the introduction and before the results? This is the conventional format for reporting studies in journals.

Response 12. The reviewer is correct in his assessment. However according to the “Instructions for authors”, the order to publish in this journal is:

    1. Introduction
    2. Results
    3. Discussion
    4. Materials and Methods
    5. Conclusions

In this sense, there is an error in the manuscript and the “Conclusions” section is moved after the “Material and Methods” section